# Periodontal Therapy Using Bioactive Glasses: A Review

John W. Nicholson [1,2]

1   Dental Physical Science Unit, Institute of Dentistry, Queen Mary University of London, Mile End Road, London E1 2NS, UK; j.nicholson@qmul.ac.uk or john.nicholson@bluefieldcentre.co.uk
2   Bluefield Centre for Biomaterials Ltd., Kemp House, 152 City Road, London EC1V 2NX, UK

**Abstract:** This paper reviews the use of bioactive glasses as materials for periodontal repair. Periodontal disease causes bone loss, resulting in tooth loosening and eventual tooth loss. However, it can be reversed using bioactive glass, typically the original 45S5 formulation (Bioglass®) at the defect site. This is done either by plcing bioactive glass granules or a bioactive glass putty at the defect. This stimulates bone repair and causes the defect to disappear. Another use of bioactive glass in periodontics is to repair so-called furcation defects, i.e., bone loss due to infection at the intersection of the roots in multi-rooted teeth. This treatment also gives good clinical outcomes. Finally, bioactive glass has been used to improve outcomes with metallic implants. This involves either placing bioactive glass granules into the defect prior to inserting the metal implant, or coating the implant with bioactive glass to improve the likelihood of osseointegration. This needs the glass to be formulated so that it does not crack or debond from the metal. This approach has been very successful, and bioactive glass coatings perform better than those made from hydroxyapatite.

**Keywords:** bioactive glass; Bioglass®; periodontology; fircation defects; bone bonding; bone putty; coatings; clinical outcomes

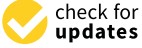



## 1. Introduction

Periodontology is the branch of dentistry that deals with the supporting structures of the teeth. It aims to maintain these structures in good health and free from disease and damage [1]. In humans, the teeth are supported by the periodontium, a complex structure embedded in alveolar bone and occurring in either the maxilla or the mandible. It includes a layer of cementum and the periodontal ligament. The latter is slightly flexible and also tough and these two features allow it to cushion the teeth as they perform their biological functions of biting, tearing and chewing. Covering the alveolar bone is the gingiva (gums), a soft tissue that is well supplied with blood vessels and which, in healthy patients, is pink in colour.

Perodontitis is an inflammatory condition that affects the supporting structures of the teeth, both the soft and hard tissues [2,3]. It is caused by specific oral bacteria that induce a host-mediated inflammatory response that results in loss of periodontal attachment [4]. It is a common chronic disease and, for example, is experienced by an estimated 70% of people aged 65 and over [3]. The World Health Organization has estimated that somewhere in the region of 80% of adults worldwide suffer from gingivitis [2] and 10–15% suffer from severe periodonitis [3]. The latter condition is the major cause of tooth loss in adults [5].

Three forms of periodontitis have been identified [6], namely necrotizing periodontitis, aggressive periodontitis and periodontitis as a manifestation of systemic disease. Whatever the form, periodontitis typically follows four stage of severity, as defined in Table 1.

The earliest stage of gum disease is known as gingivitis, and is caused by the soft supporting tissue being attacked by oral bacteria [1,3,7]. The specific species involved are *Porphyromonas gingivalis*, *Tannarella forsythia* and *T. denticola.* Infection by these organisms causes inflammation and reddening of the gums, in which condition, they bleed readily. This is only a mild infection, and it can be reversed by improving the oral hygiene, in

particular by brushing the teeth and gums at least twice a day and cleaning between the teeth with dental floss. Additional professional cleaning by a dental hygienist is also beneficial, involving as it does the removal of the hardened plaque substance known as tartar or calculus. Left undisturbed, this substance will act as a trap for bacteria and become the focus for continued gingival infection.

**Table 1.** Stages in the progression of periodontitis [6].

| Stage | Condition | Comments |
|-------|-----------|----------|
| Stage I | Borderline between gingivitis and periodontitis | Some loss of attachment. Diagnosis may be challenging in general dental practice |
| Stage II | Established periodontitis | Compromised support for tooth. Responds to relatively simple management. |
| Stage III | Significant damage to attachment with some tooth loss. | Deep periodontal lesions and furcation Involvement. Does not require complex to restore function. |
| Stage IV | Considerable damage to attachment with significant tooth loss. | Loss of masticatory function. Deep Periodontal lesions and tooth hypermobility. Case management requires stabilization/ Restoration of masticatory function. |

Gingivitis does not result in tooth loss [7]. This is because it infects only the soft tissue of the gingiva, and not the alveolar bone. However, left untreated it may develop further into the more serious condition of periodontitis. When this happens, the soft gingival tissue retreats from the base of the tooth and leaves spaces known as pockets. Bacteria from the oral plaque can then lodge in these pockets, leading to serious infection. Once this occurs, the body's immune system becomes involved, and this initiates processes that cause the bone and connective tissue below the gum line to break down. Infecting bacteria contribute to this breakdown by producing toxins, and the overall effect is that the loss of bone and connective tissue accelerates. Unless this is treated, it will eventually result in the destruction of the supporting alveolar bone, and cause the teeth to become so loose that they are eventually lost altogether.

Periodontal diseases generally do not develop until patients are at least in their thirties [1,8]. Men are more susceptible than women for reasons that have not been identified, and smokers are more susceptible than non-smokers [2]. The most obvious signs of periodontal diseases are red and swollen gums, which bleed readily. Other signs and symptoms include halitosis, painful chewing and receding gums [7].

## 2. Conventional Treatment

Clinical treatments for periodontal disease may be non-surgical ad involve techniques such as combined scaling and root planning, or surgical, involving open flap debridement [9]. Scaling and root planning aim to clean below the gum line by removing remove plaque and tartar from the pockets around the teeth. Treatment has two parts, first scaling, then planning. Scaling involves the use of hand instruments to remove plaque and tartar from the subgingival region by scraping. Planing is the step in which the roots of the teeth are smoothed and contoured, so that the soft tissue of the gingiva will re-attach. These treatments are followed up with improved oral hygiene, involving twice-daily toothbrushing and the regular use of dental floss. Medication, e.g., with doxycycline, may also be used to stop the newly cleaned pockets from becoming re-infected.

Non-surgical therapy may not eliminate the problems of the patient [9] and when this is the case, periodontal surgery may be necessary. Such surgery should only be considered when non-surgical treatment has failed to solve the problem. It involves treatment of recessions, ridge augmentation, and may also extend to peri-implant soft tissue replacement, autologous tissue grafts and placement of substitutes for soft tissue [10].

Full consideration of surgical approaches to periodontology is beyond the scope of the present article, but involves the use of appropriate flap design.

The most commonly used flap design is the coronally advanced flap [10]. However, specific clinical conditions may require the use of the laterally positioned flap. Alternatively, where there are multiple recessions, the surgical approach is guided by their number, depth and location. Where grafts are employed, the tunnel flap is the technique of choice. Full details of these surgical approached may be found in the appropriate literature [9,10]. All of these surgical processes are known to be clinically effective [11]. They aim to remove the infection and decrease the size of the periodontal pockets, making subsequent cleaning easier.

In addition, modern periodontology may make use of a number of additional surgical procedures, of which the simplest is to implant a finely-divided bioactive ceramic such as Bioglass® or hydroxyapatite which is able to promote regrowth of the alveolar bone [12]. Alternatively, implants may be placed to provide artificial anchors for teeth lost because of periodontal disease. Using implants needs care in selection of patients. Specifically, because periodontal diseases arise from poor oral health, this must be addressed if the implant is to be successful. As well as that, any alveolar bone loss must be replaced prior to the placement of implants.

Several bioactive materials are available for use in these procedures. The present review concentrates on bioactive glass for these roles, but other inorganic materials (hydroxyapatite and other synthetic calcium phosphate ceramics) have also been used with success. These are considered briefly in Section 3. Polymeric materials have also been used, mainly to fabricate tissue engineering scaffolds for periodontal repair [12–14]. Much os this has been experimental and involved animal models but results have been promising [12]. These scaffolds have been made from either natural polymers such as chitosan of collagen, or synthetic polymers, such as polylactide materials [13–15]. Natrual polymers, such as chitosan-alginate complex, have been used as drug-delivery devices for molecules active against the micro-organisms that cause periodontitis [16,17]. However, materials of this type deal with only one aspect of periodontal diseases, namely the bacterial infection, and do not address the structural damage to the alveolar bone or periodontal ligament.

A further clinical problem that is encountered in periodontology is the occurrence of furcation defects. These defects occur when periodontal disease develops in multi-rooted teeth at the points where the roots of the tooth branch [18]. If untreated, furcation defects result in loss of alveolar bone at the branching point of the tooth root. Once it has begun, the subsequent destruction of the bone occurs rapidly as plaque, tartar and bacteria move into the space at the furcation. Eliminating or reducing plaque in this region is impossible, which means that infection remains and the disease progresses rapidly to the state at which substantial bone loss occurs.

These furcation defects are likely to lead to tooth loss unless they are treated [19]. However, treatment can be difficult, because access to the diseased area by the clinician is difficult, and the anatomy prevents visual access to the affected region. Specially shaped scalers have been developed to scrape root surfaces clean around furcation defects and ultrasonic scalers can also be used to loosen debris, which can then be flushed away by a water jet. Sterile conditions can be maintained using antimicrobial compounds in the area in question. Bioactive materials, such as Bioglass®, can then be placed in the affected region in an appropriate carrier to promote deposition of bone around the affected tooth.

## 3. Materials for Periodontal Therapy

A variety of materials, both synthetic and natural, are available for use in periodontal repair [20]. There is no perfect material for bone reconstruction, and bioactive glass is one option among many [21,22]. Before focussing on bioactive glass, it is worth considering briefly the alternative materials available.

In all cases, the materials are used for regeneration of periodontal tissues adversely affected by periodontal disease. "Regeneration" is the term applied to the complete re-establishment of the periodontal tissues, including the alveolar bone, periodontal ligament

and cementum [21]. In fact, the main goal of periodontal treatment is to re-establish a complete functioning structure that includes new cementum, fibres of periodontal ligament, and a full connection to complete alveolar bone [23–25]. Complete functioning involves formation of fibres of the periodontal ligament that are properly aligned with respect to the cementum and the alveolar bone. This process may involve so-called guided tissue regeneration [14].

Some of the synthetic materials for periodontal therapy are shown in Table 2. These will be considered briefly this this section.

**Table 2.** Materials other than bioactive glass used in periodontal therapy.

| Material | Comments |
| --- | --- |
| Hydroxyapatite | Similar composition to bone but clinical results are mixed. |
| Tricalcium phosphate | Bioactive, resorbable. Clinical outcomes good. |
| Biphasic calcium phosphate | Biodegradable. Good clinical outcomes. |
| Calcium sulfate | Used as a barrier. Clinical outcomes improved by mixing with bone. |
| Degradable polymer systems | Examples: chitosan, polylactic acid. Used for drug delivery. |

Natural polymers such as chitosan have been used to treat periodontitis, with their main function being as delivery vehicles for biologically active molecules such as DNA [25] and growth factors [26,27]. Synthetic polymers (polylactic acid or lactic acid-glycolic acid copolymers) may be used to create artificial membranes able to be resorbed [27,28]. These materials allow the natural periodontal ligament to be regenerated, but no additional surgery is needed, as the polymers degrade in situ [29].

As Table 2 shows, the other materials used have been various types of inorganic material, either calcium phosphate or calcium sulfate. The calcium phosphates have particularly properties and because their composition resembles that of bone mineral, they are bioactive. This means they promote cellular function and cause healing of the bone defects. Results vary between materials, depending on their composition, state of division and porosity.

Hydroxyapatite, both natural and synthetic, is a type of calcium phosphate that has been used extensively in surgery involving bone augmentation and regeneration [12]. Its composition and structure are similar to the natural hydroxyapatite found in bones [29] as a result of which it will bond directly to bone when implanted [30]. Applications of hydroxyapatite are somewhat limited because the surface properties vary and this means that cells respond to it inconsistently [31,32]. As a result, there may be only limited bone regeneration with this material [12].

Tricalcium phosphate, TCP, can be used as an alternative to hydroxyapatite. It exists in two phases, $\alpha$ and $\beta$, of which $\beta$-TCP is particularly biocompatible in contact with bone. It has also been shown to be osteoconductive in animal and human studies [12]. Osteoconductivity is defined as the ability to allow bone to form on a surface by serving as a scaffold or as a template [33], and is an important property of materials used in periodontology. Clinical studies using granular $\beta$-TCP in the repair of periodontal defects have shown that pocket depths decrease markedly and gingival cell attachment increases within six months of placement [33,34]. Unfortunately, $\beta$-TCP does not appear to stimulate regeneration of cementum, periodontal ligament or alveolar bone [35,36], so its use in periodontology is limited.

Calcium sulfate has been used with some success in periodontal repair, mainly as a barrier in association with other graft materials [37,38]. In this way, it can help promote periodontal regeneration [39]. However, for the best clinical outcomes, calcium sulfate has to be mixed with another material, such as demineralised bone matrix, to improve its bioactivity and ability to promote bone regeneration. [40,41]

Synthetic materials are not the only materials that can be used in periodontology. Natural materials, i.e., bone of various types, can be used as well [21]. These types are as follows:

(i)   Autologous bone. This is the name given to bone that is harvested from the patient and used elsewhere in their own body to repair some sort of defect. It is considered to be the "gold standard" material for use in bone repair.

(ii)  Allografts. The term which is applied to the situation where bone is obtained from donors other than the patient themselves, either living donors or cadavers. This bone is typically freeze-dried and possibly demineralized prior to use, mainly to avoid transmission of infection.

(iii) Xenograpfts, where animal bone (bovine, porcine or equine) is used to augment the natural bone of the patient. This is fairly rare in periodontics, but has been used [21].

## 4. Bioactive Glass Materials

The field of bioceramics effectively began with the discovery of a glass capable of bonding to living bone. This material had the composition given in Table 3 [42], and was later termed 45S5 to give an indication of its composition (45% silica, 5% phosphate). It is also known commercially as Bioglass®, and has the properties given in Table 4 [43]. It was designed deliberately by Hench to be degradable [40], and the ability to bond strongly to bone was a later discovery [43]. This was a distinct advance on implantable materials then known, which were designed to be inert towards biological tissues. Such bio-inert materials generate fibrous capsule on implantation, which means that they do not form a stable interface or bond directly to the tissues.

**Table 3.** Composition of Bioglass® [38].

| Component | Proportion/mol% |
|---|---|
| $SiO_2$ | 46.1 |
| $Na_2O$ | 24.4 |
| CaO | 26.9 |
| $P_2O_5$ | 2.6 |

**Table 4.** Properties of Bioglass® [43].

| Property | Value |
|---|---|
| Density | 2.7 g cm$^{-3}$ |
| Network connectivity | 2.12 |
| Glass transition temperature | 538 °C |
| Onset of crystallisation | 677 °C |
| Coefficient of thermal expansion | $15.1 \times 10^8$/°C |
| Young's modulus | 35 MPa |

As well as bonding rapidly to bone, Bioglass® is also able to stimulate bone regeneration [42,44]. The mechanism for this has been established and occurs as follows. There is an initial dissolution of the outer layer of the glass, causing a build-up of beneficial ions in the interfacial region [45]. These ions, notably calcium and silica, cause an amorphous layer of carbonated hydroxyapatite (hydroxycarbonate apatite, HCA) to precipitate onto the surface of the glass. These ions also stimulate osteogenic cells to produce bone matrix, which contains HCA as its inorganic component [46]. Overall, the bioactivity of Bioglass® results from its rapid dissolution and the associated ready deposition of the HCA layer. The deposition of the amorphous HCA layer activates osteoblasts so that they promote the growth of new bone [47]. The mechanism is known as graft-bone bonding, and arises from the ready degradation of the Bioglass® and the specific ions that are released [48,49].

There has been a considerable volume of research on bioactive glasses since the original 45S5 glass formulation was reported, with numerous alternative formulations, including some containing borate or phosphate. However, the biological properties of 45S5 remain among the best discovered [42].

The first commercial product for periodontology that used Bioglass® was a granular substance with the proprietary name PerioGlas®. This product, which is now sold by NovaBone Products LLC of Alachua, Florida, became available in 1993, the year it received FDA approval. It's European approval, the CE mark, was granted in 1997. PerioGlas® contained glass particles in the range 90–710 µm and was designed to be pushed into place within periodontal defects during surgery. It more or less remained in place as a result of the firmness with which it was packed. When used in this way, the material has had considerable clinical success. It stimulated bone growth to such an extent that the anchorage of an affected tooth improved substantially, and tooth loosening was reversed. PerioGlas® particles have also be used to improve alveolar bone quality for titanium implants. In this case, the resulting anchorage is able to maintain support for the implants over long periods of time [50].

Since the launch of PerioGlas®, several other brands of material have become available, all aimed partly at least at being used for periodontal repair [51]. These are listed in Table 5. One brand of interest is the Consil® products, produced for a variety of functions in pets (cats and dogs), including orthopaedics as well as dental treatment. These are designed for use by veterinarians, and have shown similar outcomes to bioactive glass materials used in human patients [52].

**Table 5.** Bioactive glass products for periodontal application.

| Materials | Producer | Comments |
|---|---|---|
| PerioGlas | Ionion Oy, Tempere, Finland | |
| PerioGlas PerioGlas Bonegraft Novabone Dental Morsels Novabone BBG | Novabone Products, Jacksonville, Florida, USA | |
| PerioGlas Plus Novabone Dental Putty PerioGlas Dental Putty | | Contain water-soluble binders (glycerin, gelatin) |
| Unigraft Ossiform | Unicare Biomedical Laguna Hills, California, USA | |
| Bonalive Bonalive Putty | Bonalive Biomaterials, Turku, Finland | |
| Consil Dental (Additionally, Consil Orthopedics Consil Putty) | Nutramax Laboratories, Edgewood, Maryland, USA | For use in cats and dogs |

Bioactive glasses possess a number of advantages for use in periodontics. They have high biocompatibility in contact with bone and they promote bone growth, partly because their chemical composition is similar to that of bone, and also because they stimulate osteoblast activity in situ as a result of their slow dissolution [53]. However, they also have disadvantages. They are brittle and have low ductility [53]. As well as that, in granular form they are difficult to place with confidence. The latter feature has been addressed by the formulation of putties that incorporate bioactive glass granules in a substance that has sufficient integrity that it can be pushed into place and remain there during periodontal surgery. These putties are considered in more detail in Section 6 (b) of the present article.

## 5. Bone Bonding by Bioactive Glass

Bioactive glass works in the treatment of periodontal diseases by stimulating new bone growth. The resulting new bone replaces the bone destroyed by periodontal disease

and reverses tooth loosening. This means that the affected teeth are no longer in danger of being lost. This bone-bonding will now be considered in detail.

Both bone regeneration and bone bonding are complex processes, with the full details of the mechanism not being known [42]. Some aspects, though, have been established, especially those in the early stages.

The strong mechanical bond between bone and the bioactive glass component is the result of the layer of HCA forming. This inorganic layer interacts with collagen fibrils from the diseased bone to form a functioning bone structure [54]. After formation of the initial layer of HCA, protein molecules adsorb onto the surface, and this creates an environment to which collagen fibrils move and become incorporated. Next, bone progenitor cells become attached, followed by cell differentiation. These cells go on to excrete bone extracellular matrix followed by mineralization and the development of mature bone [41]. The details of the later steps are not clear, but it is known that the HCA is an appropriate surface for osteogenic cells to attach and proliferate.

The formation of the HCA layer occurs in five steps. These steps can occur in both natural body fluids in vivo and in simulated body fluid in vitro [55,56] and are as follows:

(i)   Formation of silanol groups on the glass surface via a cation exchange process: $-Si-O^-Na^+ + H^+ \rightarrow -Si-OH + Na^+$ This removes protons from solution, increasing the surrounding pH.

(ii)  The increased local pH has an excess of $OH^-$ ions in the solution surrounding the glass particles. These ions react with -Si-O-Si- units in the glass surface to form $Si(OH)_4$, which dissolves in the surrounding solution.

(iii) The surface -Si-OH groups undergo a condensation reaction to form -Si-O-Si- units in a pre-polymerization process.

(iv)  $Ca^{2+}$ and $PO_4^{3-}$ groups within the glass migrate to the surface and form amorphous calcium phosphate, nucleating onto the -Si-OH groups on the glass surface [57,58].

(v)   Hydroxide and carbonate ions present in the surrounding solution become incorporated into the surface layer. Subsequent interaction with the calcium and phosphate ions leads to the formation of HCA [59].

Once the HCA is formed, there are further steps leading to the attachment of bone. Detailed understanding of these steps is lacking, but some aspects are known [42]. These are:

(i)   Proteins adsorb onto the HCA surface.

(ii)  Cells attach to the protein layer, and as they do so, they go on to differentiate.

(iii) The population of differentiated cells produce bone matrix. This eventually forms fully functioning bone that is strongly bonded to the glass surface.

These reactions show that the glass surfaces have substantial biological activity. Human osteoblasts can be grown on bioactive glass and are found to produce extracellular matrix, ECM, which goes on to mineralize and form nodules of bone [60–62]. The calcium ions and the soluble silica species that dissolve from the bioactive glass stimulate cell division in osteoblasts, and also cause these cells to produce growth factors and ECM proteins [43].

The rate at which the various species dissolve from the bioactive glass surface determines the success of these processes. Dissolution must be fast enough to raise the concentration of ions sufficiently stimulate the activity of osteoblasts cells but not so fast that toxic levels build up [31]. Dissolution behaviour is controlled by the composition and structure of the glass, particularly the ratio of CaO to $P_2O_5$. This ratio which is relatively high in 45S5, hence its ready dissolution into physiological fluids. This means that this composition produces the necessary concentrations of key ions at locations where they can cause precipitation of HCA and eventually develop a population of proliferating and differentiating cells [60].

The high rate of dissolution of bioactive glasses is responsible for another biologically useful effect, i.e., raising the pH around the implanted glass particles. This pH change confers anti-microbial properties [63]. It has been shown to be effective against both the

bacteria responsible for periodontal disease, and those that cause caries [64]. Because of this, bioactive glasses tend to reduce the incidence of post-operative infections, which is an advantage in bone reconstruction surgery.

## 6. Bioactive Glasses in the Treatment of Periodontal Damage

(a)     *As Granular Fillers*

The structures supporting the teeth are complex. They consist of the hard tissues trabecular bone and cortical bone, and soft tissues, mainly bone marrow and the periodontal ligament [60]. Periodontal disease affects all of these tissues, and various approaches have been taken to treat these effects, with the use of granular bioactive glass being particularly effective.

Bone loss from the alveolar region is the defining outcome of periodontal disease, and is caused by progression of bacterial decay from the initial infection. To the stage at which the quality of bone is severely compromised. Other tissues may be affected adversely, but the loss of bone is the most serious. This is because it leads to loosening of the teeth and their eventual loss [63].

The extent of bone loss is assessed clinically using radiographs. This enables the clinician to diagnose the condition and carry out treatment planning aimed at rectifying the damage [62]. Bone augmentation promoted by bioactive glass is widely used because this material is able to cause specific responses in the remaining healthy cells of the periodontium. It promotes osteogenesis, thereby stimulating the formation of new bone [65]. It may also act as a barrier to epithelial cells, stopping them from growing downwards, and thus acid as a guide for tissue growth so that it occurs in the correct biological orientation. The high pH generated in the fluids surrounding the glass particles has antimicrobial effects, which are also beneficial in vivo [66].

Bioactive glass stimulates the development of fully functioning bone, not just deposition of the mineral phase. If the particles are porous, optimal levels of vascularisation can occur. These glass particles are easy to manipulate in the clinic and they are haemostatic, i.e., they stop the bleeding. This gives a clear working area for the clinician and also improves ease of use [60].

Bioactive glass in the form of granules has been very successful in the treatment of periodontal disease [47,67]. Clinical treatment for this condition has three desirable outcomes, and these are (i) reduction in gingival pocket depths at each tooth, (ii) increase in the clinical attachment of the soft tissue to the supporting bone, and (iii) improvement in the quantity and quality of the alveolar bone surrounding the tooth socket. Bioactive glass provides all three [67–70].

There have been numerous clinical studies published that demonstrate how bioactive glass is effective [69–72] and, in periodontology, leads to all three desirable outcomes. In one study, Nevins et al. reported on the treatment of intrabony defects around five teeth with bioactive glass [73]. Using clinical radiographic measurements, they were able to show that, six months after treatment, the depth of the periodontal pockets had decreased by a mean of 2.7 mm. Meanwhile, the gingival attachment to the alveolar bone around the socket had increased by a mean of 2.2 mm. Additional histological study showed that there was new cementum and new connective tissue formed in the region of the implant in one of the cases, which was a particularly striking result [47]. In the other four cases, healing was less extensive, but still showed distinct bone bonding and also the formation of new junctional epithelium. The repairs in this study were of small infrabony defects, and bioactive glass has been found to be particularly suitable for use in this type of condition [73].

The reduction in pocket depth within a few months of placing bioactive glass has been observed in many studies [67,73–75]. This is usually associated with a significant increase in clinical attachment of the gingival tissue. However, this attachment has been found to vary somewhat, and may range from so small that it not statistically significant [73,76] to the very considerable [74]. These differences may be influenced by the extent to which the disease has progressed when the treatment was carried out. With more advanced disease,

gingival detachment is greater, and repair at the treatment site is probably faster than at sites where the disease is less advanced [74].

There have been a sufficient number of randomized control clinical trials of bioactive glass granules in periodontal repair that the effectiveness of the material is well established [66]. A meta-analysis of such trials reported in 2012 that, in the clinical trials to date, there had been a mean reduction in pocket depth of 0.72 mm and a mean increase in clinical attachment of 1.18 mm [66]. Taken overall, these studies all point to the same thing, namely that repairing periodontal defects with bioactive glass granules is a beneficial intervention. Moreover, these good outcomes have been achieved with no reports of adverse effects such as allergies or other immunological responses [74], and no reports of abscess formation or of rejection of the material [66]. As well as reducing pocket depth and increasing attachment, periodontal therapy should ideally lead to regeneration of the alveolar bone and to the defects disappearing as they become filled with new bone. This has been widely observed with bioactive glass granules [70,73,76]. Post-operative healing after placement of bioactive glass granules occurs quickly [70,73,77,78] and, overall, bioactive glass has been shown to be highly efficacious in this application [79].

As well as using bioactive glass on its own to promote bone growth in the periodontium, it has been used with resorbable and non-resorbable membranes [80]. These membranes' function is to prevent the migration of epithelial cells into the underlying graft site. This means that other cell types can become attached at the graft site. They then populate the defect and cause the bone to grow. This is called guided tissue regeneration [79], and is enhanced by the addition of growth factors to the glass particles [79]. Unfortunately, in clinical conditions, the addition of such additional biomolecules give very variable outcomes. Success depends on several factors, including the specific tooth involved, the health of the bone at the defect site, and the quality of the oral health of the patient. The use of blends of bioactive glass with biomolecules is the subject of current research, and the aim is to fabricate reliable blends that can provide better regenerative outcomes than bioactive glass alone.

(b)    *Putties containing bioactive glass particles*

An alternative way of presenting the bioactive glass for periodontal repair is as the filler in a putty [81]. A typical commercial material of this type is NovaBone Putty and it is made from a bioactive glass powder blended with a polyethylene glycol and glycerin binder. This blend needs no preparation in the clinic, but can simply be placed directly at the site of bone loss. It does not undergo hardening, but remains deformable. However, with time, the binder fluid may be resorbed, leaving the bioactive glass particles behind. NovaBone Putty was approved for clinical use in the US in 2006 and in Europe in 2007, and is a better way of presenting bioactive glass than simply using the unblended glass granules.

The ultimate goal of employing this type of bioactive glass putty is the same as for using glass particles, namely the restoration of the periodontal structures damaged by disease [77]. The putty is easier to press into place that bioactive glass particles, and is also more likely to remain in place.

Studies have confirmed that this type of putty can provide reliable and acceptable clinical results [81]. One study showed that, six months after placement, the probing depth of the periodontal pockets had become reduced by a mean of 4.2 mm. Both soft and hard tissues respond positively to the presence of the putty, and within six months the inflammation due to periodontal disease had completely resolved. Results resembled those with particulate bioactive glass powders [68,76,82,83], even to the extent to which the size of periodontal pockets decreased. However, the putty was easier to use and more reliable, as well as being well tolerated by patients. These studies show that the putty is an acceptable way of introducing bioactive glass under surgical conditions, and that it gives successful clinical outcomes.

(c) *Treatments for furcation defects*

An important problem in clinical periodontology is the occurrence of furcations in multi-rooted teeth that act as sites promoting periodontal disease [84,85]. The presence of these furcations can cause disease to develop because they are readily infected, and infection is followed by inflammation, and eventual bone resorption. These defects typically result in tooth loss, and tooth loss in these circumstances is more common than in teeth without furcation defects [86].

Clinically, treatment of these defects is aimed at the regeneration of natural tissues at the affected sites. Regeneration can be achieved in various ways, including bone grafting, but the use of bioactive glasses has been a particularly successful approach [87]. Bioactive glasses are able to promote osteogenesis and cementogenesis at affected sites, and are also able to stimulate the development of fully functional periodontal ligament [88].

Bioactive glass is easy to place at the furcation defect and it remains in place even when suction is applied close to the site [86]. This is because it forms a cohesive mass with saline or even blood, and this mass is retained at the site rather flowing away, even when there is bleeding. Bioactive glass is haemostatic, which means it rapidly forms a blood clot at the furcation defect, and in this way initiates the healing process [86].

In one study of bioactive glass for the treatment of furcation defects, glass particles were found to remain in place for ten days after surgery [86]. In most of the patients, the overlying mucoperiosteal flaps were healthy at this time, and the whole site was well underway to healing. The reported success rate was 94%, and what failures there were, were all the result of infection and inflammation. The failures were all due to poor oral hygiene by specific patients, that meant that proper healing conditions were not maintained [89]. This demonstrated the importance of patient compliance in the success of these procedures, a feature that needs to be considered in selecting patients for this treatment.

In most cases, the use of bioactive glass led to successful outcomes, with the overlying gingival tissues being able to tolerate the presence of the glass particles. Soft gingival tissue generally healed well and gave stable wounds and good regeneration of the overall periodontal structures [86].

When sites were examined radiologically after six months, there had been significant replacement of the lost bone at the defect site, with the new bone being of good density [86]. Other studies of bioactive glass in direct contact with human bone show that the particles begin to disappear at about 4 months and are completely resorbed at 16 months [89]. Clinical studies of the repair of furcation defects give similar results and show that bioactive glass is able to promote regeneration of bone at defect sites. This procedure allows teeth affected by furcation defects to be saved, and the periodontium to be restored to full function, rather than being lost.

(d) *Coatings for implants*

Bioactive glass has also been used to coat implants used in dentistry [90]. Dental implants are typically made of the titanium alloy Ti-6Al-4V and are used to support ceramic crowns or a group of mainly ceramic prosthetic teeth. This alloy shows good biocompatibility in contact with bone. In this treatment, patients must have high standards of oral hygiene and be non-smokers. They must also comply carefully with clinical instructions following the placement of these implants, because long times are needed for the implant to become fully integrated. These are typically between three and six months for both the mandible and the maxilla [91].

Currently there is interest in modifying titanium alloy surfaces to improve their osseointegration. One approach is to increase the overall surface area of the implant using grit-blasting or acid-etching. Another is to coat the surface with a bioactive material.

Such coatings have been made with synthetic hydroxyapatite, but success with them has been limited [92,93]. This is because a gap develops in the interfacial zone between the implant and the bone due to the release of high local concentrations of ions from the coating [94]. These high concentrations have an adverse effect on the bone, causing it to

be lost from the region adjacent to the implant. The resulting gap causes loosening of the implant, leading to eventual failure.

On the other hand, when bioactive glass is used to coat metallic dental implants, results are much better [95]. This is because bioactive glass is able to form a strong chemical bond to living bone [96]. Its greater bioactivity promotes the growth of bone in the interfacial region better than hydroxyapatite, and there are no problems of bone resorption. Consequently, the implant becomes stabilized much faster, and the bone that forms is attached firmly to the surface of the metal implant.

Forming a reliable coating from a glass is difficult, because stresses arise at the interface between the metal and the coating due to their different thermal properties [91]. As a result, the glass coatings may crack and debond [96]. To prevent this, the fabrication conditions have to be carefully chosen [94]. One approach has been to change the composition of the bioactive glass, specifically by replacing $CaO$ with $MgO$ and $Na_2O$ with $K_2O$. This alters the thermal expansion properties of the glass, so that the coefficient of thermal expansion more closely matches that of the metal alloy substrate [97].

The subject of fabricating glass and using it at an appropriate thickness on dental implants is complicated and there have been numerous studies published [92,96,98,99]. Animal studies have shown that this type of coated implant performs well in living bone, becoming strongly integrated into the host bone with no sign of fibrous capsule formation. Additionally, the resulting bone density is much greater than that which forms around uncoated implants [95,100].

One important issue with dental implants is that they need to penetrate the soft tissues and this involves a region on the exposed part that hosts various microorganisms [61]. This may cause infections to develop at the penetration site. However, if the soft tissues grow to form healthy zones around the emerging implant, an attachment forms which acts as a biological seal [75]. This seal keeps the bone isolated from the mouth and prevents the spread of infection. Because bioactive glass is active against infectious bacteria [62], it can further reduce the chance of infection at the penetration site, and improve the chances of clinical success.

So far, there have been only a few detailed studies of the performance of glass coated implants. However, results generally confirm that these implants perform well [99,101] and typically after 12 months, they have sound bone growing right up against them and strong biological fixation. Longer-term clinical studies are needed to confirm these findings and to show whether bioactive glass coatings really do improve the in vivo performance of metal implants [92].

(e)     *Bone augmentation prior to the use of implants*

Seeing that bioactive glass is capable of promoting bone regeneration, it has the potential to be used to stimulate bone growth around metal implants in patients who have had severe periodontitis and lost teeth as a result. This is not completely straightforward, because periodontitis is usually associated with poor oral hygiene and possibly smoking by patients. Using implants in such patients is usually considered contraindicated [102].

However, in spite of this, bioactive glass has been used in these circumstances this way and given successful outcomes [103]. In one study, three patients were given the bioactive glass PerioGlas® at extraction sites, after which they each received a titanium alloy dental implant. The aim was to form new bone in which the implant could be reliably fixed. This was duly achieved, and by six months, when bone biopsies were performed, new bone had successfully developed. At the same time, the glass granules had almost completely degraded. A two-year follow up showed that all the implants were stable, and that satisfactory repairs had been made in each case [101]. The fact that such unpromising circumstances led to clinical success shows how remarkably bioactive this glass composition is.

(f)     *Future directions*

Clinical results for bioactive glass show them to perform well in the repair of lesions caused by periodonitis [67], so there is little scope for improvement here. The original

bioglass formulation, so-called 45S5 has excellent biological properties, which have not been improved upon. This formulation gives very good clinical outcomes when used in periodontal repair. It is likely that future studies will improve our understanding of its biological properties, especially its interactions with living cells, but this is unlikely to result in improvements in its clinical performance.

The same is not true of bioactive glasses for coating applications on implants. Here, there is room for improvement. Coatings need to have greater ability to be retained in the surface region of the implant, and not debond due to mismatch in modulus between the glass and the metal substrate. New coating techniques will have to be explored with the aim of enhancing clinical performance, particularly in medically compromised patients. New bioactive glass compositions may be needed in order to create materials with higher strengths, better mechanical properties, and increased bioactivity.

## 7. Conclusions

The discovery of the bioactivity of Bioglass®, also known as 4S5S in 1969, led to the development of a number of bioactive glass materials for periodontal therapy, with the first being PerioGlas®. Since the original particulate products became available, putties containing similar bioactive glasses have been developed with the glass particles dispersed in viscous liquid carriers. These have been found to be easier to use place than the glass particles alone, but to have similar clinical benefits.

These glass-based materials give satisfactory clinical outcomes. Periodontal pockets become less deep, gingival attachment increases, and defects become filled with fully functioning natural bone. These represent a reversal of the damage done by periodontal disease, and improve the chances of retaining teeth in the affected region.

Bioactive glass granules and putties have also been used for repair of furcation defects, also with good clinical outcomes. Putties give better results than glass particles alone, because they are easier to retain at the defect site.

Coatings of bioactive glass may improve the interaction of metal implants with bone, thereby increasing the strength of the implant anchorage. Alternatively, bioactive glass particles may be placed into bone prior to the implant, and this may also help with the fixation of the implant.

In these various way, bioactive glasses have made a positive contribution to the oral health of numerous patients across the world, and there is every prospect of this continuing well into the future.

**Funding:** This research received no external funding.

**Institutional Review Board Statement:** Ethical review was not appropriate for this study, which was entirely literature-based, and involved no experimental work on humans or animals.

**Informed Consent Statement:** Not applicable.

**Data Availability Statement:** Not applicable.

**Conflicts of Interest:** The author declares no conflict of interest.

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
