# Peer review of "Periodontal Therapy Using Bioactive Glasses: A Review"

_prosthesis, doi:10.3390/prosthesis4040052_

Round 1

Reviewer 1 Report

Dear Editor, 

This is an interesting review article but there are some comments which should be considered before next step: 

- It would be better to added new studies in periodontal diseases and explain the problems very well. 

- Author should explain the mechanism of Bioactive glass in periodontal therapy

-Author should explain the current periodontal therapies and advantages of Bioactive glass and disadvantages of other materials about 500 words

-it would be better to add the future direction 

-I suggest citations to following related papers to expand the introduction and discussion sections.

Yazdanian, M., et al. Evaluation of antimicrobial and cytotoxic effects of Echinacea and Arctium extracts and Zataria essential oil

(2022) AMB Express, 12 (1), art. no. 75, .

Barzegar, P.E.F., et al. The current natural/chemical materials and innovative technologies in periodontal diseases therapy and regeneration: A narrative review (2022) Materials Today Communications, 32, art. no. 104099, .

Rajabi, Z., et al. Comparison of the Effect of Extracted Bacteriocin and Lytic Bacteriophage on the Expression of Biofilm Associated Genes in Streptococcus mutans (2022) Advances in Materials Science and Engineering, 2022, art. no. 5035280, .

Yazdanian, M., et al. Chemical Characterization and Cytotoxic/Antibacterial Effects of Nine Iranian Propolis Extracts on Human Fibroblast Cells and Oral Bacteria (2022) BioMed Research International, 2022, art. no. 6574997, .

Yazdanian, M., et al. The Potential Application of Green-Synthesized Metal Nanoparticles in Dentistry: A Comprehensive Review (2022) Bioinorganic Chemistry and Applications, 2022, art. no. 2311910, .

Yazdanian, M., et al. Decellularized and biological scaffolds in dental and craniofacial tissue engineering: a comprehensive overview

(2021) Journal of Materials Research and Technology, 15, pp. 1217-1251.

Tafazoli Moghadam, E., et al. Current natural bioactive materials in bone and tooth regeneration in dentistry: a comprehensive overview

(2021) Journal of Materials Research and Technology, 13, pp. 2078-2114.

Soudi, A., et al. Role and application of stem cells in dental regeneration: A comprehensive overview

(2021) EXCLI Journal, 20, pp. 454-489.

Motallaei, M.N., et al. Evaluation of Cytotoxic and Antimicrobial Properties of Iranian Sea Salts: An in Vitro Study

(2021) Evidence-based Complementary and Alternative Medicine, 2021, art. no. 8495596, .

Hakim, L.K., et al. Biocompatible and Biomaterials Application in Drug Delivery System in Oral Cavity

(2021) Evidence-based Complementary and Alternative Medicine, 2021, art. no. 9011226, .

Motallaei, M.N., et al. The Current Strategies in Controlling Oral Diseases by Herbal and Chemical Materials

(2021) Evidence-based Complementary and Alternative Medicine, 2021, art. no. 3423001, .

Moghadam, E.T., et al. Current herbal medicine as an alternative treatment in dentistry: In vitro, in vivo and clinical studies

(2020) European Journal of Pharmacology, 889, art. no. 173665,

Reviewer 2 Report

Dear,

This manuscript can be accepted but there are some comments:

-Authors should make a part in mechanism of Bioglass treatment. 

- Authors should prepare a table and list the new studies in related treatment of Bioglass periodontal diseases. 

- Authors should make a part and explain the role the Bioglass in periodontal diseases. 

Reviewer 3 Report

This article reviews the application of bioactive glass materials in periodontal treatment. The topic selection has certain clinical significance, but the content is scattered, the focus is not outstanding, the logic is not good.

You are advised to modify the summary. The present content is unfocused and should be revised. This PAPER SHOULD NOT ONLY FOCUS ON THE CHARACTERISTICS OF BIOACTIVE GLASS MATERIALS AND THEIR ADVANTAGES IN PERIODONTAL APPLICATIONS, BUT ALSO BE LOGICAL AND UNIVERSAL.

The first three parts of this paper introduce the background knowledge of periodontal disease and the current treatment dilemma, the length is too large, need to be condensed.

There are some inaccuracies in the description of the background of periodontal disease.

Such as:

Line 29 Periodontal complex of the periodontal "it includes a layer of bone cement and periodontal ligament" should also include alveolar bone.

The staging of periodontal disease in Table 1 is not accurate enough. It is recommended to refer to the new classification of periodontal disease in 2018

Line 57 GUN is unprofessional. It appears to be a bone defect in cement enamel under CEJ

Line 62 "Periodontal disease usually does not develop until patients are at least in their thirties. Studies show that about 48 percent of adults in the United States have chronic periodontitis, and other countries appear to have similar disease prevalence rates.

The two references here are from 1983 and 2005. Please confirm the validity of these references in light of the latest epidemiological data and correct for racial and age differences in the onset and progression of periodontal disease.

Line 69 "Either a combination of scaling and root planning or open flap debridement" Treatment of periodontal disease is not just scraping, root surface leveling and flapping, which can be summarized as non-surgical treatment and surgical treatment.

Line 80 "In contemporary dentistry, materials with the ability to regenerate alveolar bone are usually followed by placement.

Common flap surgery and regenerative surgery are different, and it is recommended to distinguish the details here.

Line 84 "However, modern periodontology may make use of some additional surgical procedures. We are puzzled that the "other surgical procedures" here are not specified in detail, and if it refers to regenerative surgery, it is suggested to be combined with the previous content.

Table 5 in Part IV lists the bioactive glasses used for periodontal treatment by different manufacturers, and it is meaningless to list only the manufacturers without comparing them with each other. It is suggested to compare the concentrations of related components to explore the effect of different concentrations on the activity of bioactive glass.

The fifth part is also very long. It is suggested that the authors list the factors affecting the "bone-binding" of bioactive glass in the section for easier understanding.

Part VI: 6. Application of bioactive glass in periodontiology

This section can be divided into five parallel sections :(a) as pellet filler (b) putty containing bioactive glass particles (c) treatment for finding defects (d) coating of the implant (e) bone increment prior to application of the implant. However, we believe that the relationship between these five parts is not parallel and there is a logical error. (a), (b) and (c) are about the forms of use of bioactive glass, and (c) and (E) are indications for the use of bioactive glass. It is recommended to rewrite this section.

The conclusion is also very long and partly inappropriate.

Line 256 "This means that the affected tooth is no longer at risk of loss. We think this statement is too absolute.

Lines 535-544 restate Part VI, repeating rather than summarizing or discussing it. It is recommended that this part be cut and refined.

Round 2

Reviewer 2 Report

Dear, 

The revised manuscript is acceptable

Author Response

Thank you for this final comment and for recommending publication.

Reviewer 3 Report

Q1. We still think the logic of this paper is not clear.

Generally speaking, the subtitles of the article can play a reminder and point out the role, of which their relationship should be directly parallel or progressive.

We have excerpted the title and subtitles of this article.

title: Periodontal therapy using bioactive glasses: A review

subtitles:

1. Introduction

2. Conventional treatment

3. Materials for periodontal therapy

4. Bioglass®

5. Bone bonding

6. Use of bioactive glasses in periodontology

7. Conclusions

First of all, we believe that Part 2 and 3 mentioned the background information of periodontal diseases and treatment. We suggest authors to combine the two parts, cut out the lengthy content and put forward the novelty and importance of biomaterial adjuvant therapy.

Then, we would like to ask the author to explain the writing ideas and internal logical relations of parts 4, 5 and 6.

We don't know if I got that right. We think that the author introduced the material in Part 4 first, then introduced the mechanism in Part 5 and then introduce its periodontal application in Part 6. If so, we don't think the subtitles could match the content.

We don't think simple words or phrases such as “Bioglass® ” and “Bone bonding” are suitable for subtitles. Besides, we think that “Use of bioactive glasses in periodontology” is too similar to the headline.

Q2. Part VI: 6. Application of bioactive glass in periodontiology
This section was divided into six parallel sections :(a) as pellet filler (b) putty containing bioactive glass particles (c) treatment for finding defects (d) coating of the implant (e) bone increment prior to application of the implant (f) Future directions

And in the cover letter, the author mentioned that “the topics covered are all relevant to a consideration of the use of bioactive glass in periodontics, so I would prefer to leave the topics covered as they are.”

We still believe that the relationships between these parts are not parallel and there is a logical error. (a), (b) and (c) are about the forms of use of bioactive glass, and (c) and (e) are indications for the use of bioactive glass.

We agreed with that “the topics covered are all relevant to a consideration of the use of bioactive glass in periodontics”, however we think that the relationships between the parts are equally important, and the text needs to be organized to be readable, not just a pile of information. Thus, we still recommend authors to rewrite this section.

Q3. Line 51: Table 1: Stages in the progression of periodontal disease [6]

Please check the Ref.6 in Line 731 “6. Tonetti, M.S.; Greenwell, H.; Kornman, K.S. Staging and grading of periodontitis:Framework and proposla of a new classification and case definition. J. Periodontol., 2018, 89, 159-172”

Please note that Periodontal disease is not equal to Periodontitis.

We think that Table1 showed the stages of periodontitis instead of “periodontal disease” and wonder that why the authors didn’t show relative information of grading of periodontitis.

Q4. Line 96: The author mentioned that “Once this occurs, the body’s immune system becomes involved, and this initiates processes that cause the bone and connective tissue below the gum line to break down.”

The word “the gum line” is ambiguous. Gingiva can be classified as free gingiva, attached gingiva, or do you mean margin gingiva?

Q5. The author clarified in the cover letter that “Almost all studies are of Perioglass, and this list appears in order to demonstrate that, as far as we know, the findings are relevant for this class of material, not one single brand. ”

However, actually they didn’t mention full manufacturer information of each product in Table 5.

For example, we didn’t find the producer information of PerioGlas Plus and wonder if it could be different from PerioGlas. However, there was no relative information found in the “Comments” area.

Q6. Line 669: The author mentioned that “It is likely that future studies will improve our understanding of its biological properties, especially its interactions with living cells, but this is unlikely to result in improvements in its clinical performance. ”

We have different opinions on this sentence.

We believe that it is important to study the effects of bioactive glass on cells, for example, similar materials may affect stem cell differentiation and provide the possibility of bone regeneration in the clinic.

In addition, bioactive glass treatment of periodontitis can also be combined with infrared excitation and photodynamic therapy, please add relevant content.

Q7. Line 683: The author divides the conclusion part into five natural paragraphs, which we think is too much and the content is scattered.

Line 684: The author mentioned that “The discovery of the bioactivity of Bioglass®, also known as 4S5S in 1969, led to the development of a number of materials for periodontal therapy, with the first being PerioGlas®.”

We don't think it is necessary to review and introduce the material background again in the section. We suggest that the authors could rewrite the conclusions and summarize the main points of the paper.

Author Response

I don't agree with most of the comments under Q1, so have made only a few minor changes.

In composing sections 4, 5 and 6, the ideas followed the order: (4) The material; (5) its key feature; (6) its specific clinical use. I have no plans to change this order or to justify it further.

I have changed the subtitles slightly, to try and address the reviewer's criticism.

Q2: I am mot prepared to change anything as I do not agree with the reveiwer's criticism (and neither does the other reviewer!).

Q3: I have changed the term to "periodontotitis". I am not prepared to make any other changes because this is not a review of periodontitis, but of the use of bioactive glass in its treatment.

Q4: I don't know why the term "gum line" is considered ambiguous because it appears in several of the references cited.  Consequently, I have not changed it.

Q6. The reviewer seems to be after information that is proprietary.  I have explained why I included information in the way I did, and am not changing it.

I appreciate that we have different opinions, but as the article has my name on it, my opinion is the one that will prevail.

Q7. I have made minor changes to the Conclusions, but since the author cannot explain clearly what he wants, this is my final attempt.